

# A series of climate oscillations around 8.2 ka BP revealed through multi-proxy speleothem records from North China

Pengzhen Duan[1], Hanying Li[2], Zhibang Ma[3], Jingyao Zhao[2], Xiyu Dong[2], Ashish Sinha[4], Peng Hu[5,6], Haiwei Zhang[2], Youfeng Ning[2], Guangyou Zhu[1], Hai Cheng[2,7,8]

[1]Research Institute of Petroleum Exploration and Development, PetroChina, Beijing, China

[2]Institute of Global Environmental Change, Xi'an Jiaotong University, Xi'an, China

[3]Key Laboratory of Cenozoic Geology and Environment, Institute of Geology and Geophysics, Chinese Academy of Sciences, Beijing, China

[4]Department of Earth Science, California State University, Dominguez Hills, Carson, USA

[5]Yunnan Key Laboratory of Meteorological Disasters and Climate Resources in the Greater Mekong Subregion, Yunnan University, Kunming 650091, China

[6]Department of Atmospheric Sciences, Yunnan University, Kunming 650500, China

[7]State Key Laboratory of Loess and Quaternary Geology, Institute of Earth Environment, Chinese Academy of Sciences, Xi'an, China

[8]Key Laboratory of Karst Dynamics, MLR, Institute of Karst Geology, CAGS, Guilin, China

*Correspondence to*: Hanying Li (hanyingli@xjtu.edu.cn) and Hai Cheng (cheng021@xjtu.edu.cn)

**Abstract.** The 8.2 ka event has been extensively investigated as a remarkable single event, but rarely considered as a part of multi-centennial climatic evolution. Here, we present absolutely dated speleothem multi-proxy records spanning 9.0–7.9 ka BP from Beijing in North China, near the northern limit of the East Asian summer monsoon (EASM) and thus sensitive to climate change, to provide evidence for the intensified multi-decadal climatic oscillations since 8.5 ka BP. Three extreme excursions characterized by inter-decadal consecutive $\delta^{18}O$ excursions exceeding $\pm 1\sigma$ are identified from 8.5 ka BP in our speleothem record. The former two are characterized by enriched $^{18}O$ at ~8.40 and 8.20 ka BP, respectively, suggesting a prolonged arid event which is supported by the positive trend in $\delta^{13}C$ values, increased trace element ratios, and lower growth rate. Following the 8.2 ka event, an excessive rebound immediately emerges in our $\delta^{18}O$ and trace element records but moderate in the $\delta^{13}C$, probably suggesting pluvial conditions and nonlinear response of the local ecosystem. Following two similar severe droughts at 8.40 and 8.20 ka BP, the different behavior of $\delta^{13}C$ suggests the recovering degree of resilient ecosystem responding to different rebounded rainfall intensity. A comparison with other high-resolution records suggests that the two droughts-one pluvial patterns between 8.5 and 8.0 ka BP are of global significance instead of a regional phenomenon, which is causally linked to the slowdown and acceleration of the Atlantic Meridional Overturning Circulation that was further dominated by the freshwater injections in the North Atlantic.



## 1 Introduction

The overall warming during 9.0–7.9 ka BP (thousand years before present, where the present is 1950 CE) was punctuated by several inter-decadal to centennial climate fluctuations in the Northern Hemisphere (NH). The 8.2 ka event, as the most prominent abrupt cold event registered in the Greenland ice core records within the Holocene (Thomas et al., 2007), has been widely revealed by a large number of marine and terrestrial archives and dated to occur between 8.3–8.0 ka BP with a duration of 150–200 years (Figure S1) (e.g., Alley et al., 1997; Thomas et al., 2007; Kobashi et al., 2007; Cheng et al., 2009; Liu et al., 2013; Morrill et al., 2013; Duan P et al., 2021).

With deeper investigation, the "cold event" at 8.2 ka BP is evidenced likely to be a part of larger "set" of cold climate anomalies between 8.6 and 8.0 ka BP (e.g., Rohling and Pälike, 2005). According to marine records, the freshwater drainage(s) of proglacial Lakes Agassiz-Ojibway (LAO) into the North Atlantic, which has commonly been thought to trigger the 8.2 ka event (e.g., Alley et al., 1997; Barber et al., 1999) through weakening the Atlantic meridional overturning circulation (AMOC) and resultant global impact, is supposed to separate into two stages (Ellison et al., 2006; Roy et al., 2011; Godbout et al., 2019, 2020) or multiple outbursts (e.g., Teller et al., 2002; Kleiven et al., 2008; Jennings et al., 2015). The first pulse of freshwater may have induced the freshening of the North Atlantic at 8.55–8.45 ka BP (Lochte et al., 2019), the abrupt sea level jump (Tornqvist and Hijma, 2012; Lawrence et al., 2016), the detrital carbonate peak at ~8.6 ka (Jennings et al., 2015), and deposition of a red-sediment bed in Hudson Strait at ~8.26–8.69 ka BP (Kerwin, 1996; Lajeunesse and St-Onge, 2008). The superimposed effect of two or more successive freshwater drainages, or probably coupled with meltwater flux from the ice sheet (Morrill et al., 2014; Matero et al., 2017), finally led to severe and dramatic cooling events in the NH (Teller et al., 2002; Ellison et al., 2006). This is consistent with the view that the 8.2 ka event commenced at ~8.5 ka BP and persisted until ~8.0 ka BP (Rohling and Pälike, 2005) with more than one multi-decadal or centennial perturbations (e.g., Daley et al., 2009; Domínguez-Villar et al., 2009; Tan et al., 2020; Duan W et al., 2021). However, some terrestrial records, such as the Greenland ice cores (Thomas et al., 2007) and European lake sediments (von Grafenstein et al., 1999; Andersen et al., 2017), only documented a remarkable climate event at ~8.2 ka BP, whereas the counterpart to the preceding perturbation is not registered.

On the other hand, the multi-decadal or centennial perturbations aforementioned trended not only to the cold and dry direction in the NH, but also extremely warm and humid condition that has been evidenced in the immediate aftermath of the 8.2 ka event (Andersen et al., 2017; Duan P et al., 2023). In particular, the post-event excessive rebound suggests a major pluvial episode prevailing across a large part of North China (Duan P et al., 2023). However, only one proxy, speleothem $\delta^{18}O$ (Duan P et al., 2023), is insufficient and thus multi-proxy evidences about the overshoot is necessary, especially from the Asian summer monsoon (ASM) domain where the climate change has a fast atmospheric teleconnection with the high-latitude North Atlantic (Cheng et al., 2020; 2022), to complement our understanding on the dynamics of rapid climatic changes, their underlying mechanisms, and the local ecosystem response.

In the context of high-emission greenhouse gas nowadays, the melted Greenland ice sheet will inject huge amount of freshwater into the North Atlantic in the next millennium, which is analogous to the sea level rising scenario during 9.0–7.9 ka BP (e.g., Aguiar et al., 2020). Therefore, it is important to elucidate the climate variations in response to the freshwater injections in the past to provide a potential analogy for future behavior, especially in North China where





the ecosystem and economic development are highly dependent on hydroclimatic changes. Importantly, our study area
is located near the northern fringe of the East Asian summer monsoon (EASM), thus sensitively responding to the
variations of EASM intensity (Duan et al., 2014; Li et al., 2017; Ma et al., 2012). Here we provide high temporal
resolution speleothem multi-proxy records, including $\delta^{18}O$, $\delta^{13}C$, Mg/Ca, Sr/Ca, and Ba/Ca, from Beijing in North
China to reconstruct the hydroclimatic variations over the Circum-Bohai Sea Region (CBSR) between 9.0–7.9 ka BP.
Two cold climate anomalous events pre- and at 8.2 ka BP, as well as post-8.2 ka rebound, are investigated to show
the general climate pattern around the abrupt cold event from its triggering, response, and ensuing feedback and further
examine the relationship between the ASM and the North Atlantic.

## 78    2 Materials and Methods

### 79    2.1 Regional settings and modern climatology

Situated at ~60 km southwest of Beijing in North China, the Huangyuan Cave (39°42′ N, 115°54′ E, altitude 610 m
above sea level) is developed in a Middle Proterozoic dolomite and adjacent to Kulishu (39°41′ N, 115°39′ E) and
Shihua (39°47′ N, 115°56′ E) Caves (Figure S1). The vegetation above the cave is dominated by secondary-growth
deciduous broadleaf trees and shrubs (Ma et al., 2012; Duan et al., 2014). According to the meteorological station's
observed data between 1998 and 2010, the average annual air temperature and precipitation in the study area are
12.2 °C and 540 mm, respectively, with cold dry winters and warm wet summers (Figure 1). The regional precipitation
is highly seasonal and mainly concentrates on the summer season. It has been demonstrated (Duan et al., 2016; Li et
al., 2017; Duan P et al., 2023) that the summer precipitation $\delta^{18}O$ ($\delta^{18}O_p$) is negatively correlated with the summer
rainfall amount over the study area and positively correlated with $\delta^{18}O_p$ over almost the entire EASM domain, the
latter of which is normally termed the EASM intensity.
Speleothem BH-2, collected from Huangyuan Cave, is ~17 cm in length and ~5 cm in width (Figure 2a). The
candlestick shape of speleothem without macroscopic bias of the growth axis signifies that it was deposited under
relatively stable conditions (Baker et al., 2007). The results for the section of 15–48 mm from the top of the sample,
corresponding to 8.38–8.06 ka BP, have been reported in previous investigation (Duan P et al., 2023). In this study,
the multi-proxy results of the entire sample are presented that spans 9.0–7.9 ka BP.

### 95    2.2 [230]Th dating, stable isotope, and trace element analysis

A total of 22 [230]Th dates (Table S1) were performed at University of Minnesota, USA, using Thermo-Finnigan
Neptune multi-collector inductively coupled plasma mass spectrometers (MC-ICP-MS, Thermo Scientific). The
methods are described in detail in Cheng et al. (2013). We followed standard chemistry procedures to separate uranium
and thorium for instrument analysis (Edwards et al., 1987). A triple-spike ([229]Th-[233]U-[236]U) isotope dilution method
was employed to correct instrumental fractionation and determine U/Th isotopic ratios and concentrations.
Uncertainties in U/Th isotopic data were calculated offline at 2σ level. Confocal Laser Fluorescent Microscopy was
used to observe clear annual bands for the section of 15 to 48 mm, each of which comprises paired light and dark
lamina, and the results have been reported in previous study (Duan P et al., 2023).





The stable oxygen and carbon isotopes ($\delta^{18}O$ and $\delta^{13}C$) of speleothem BH-2 were determined on a Thermo-Scientific
MAT-253 isotope ratio mass spectrometer equipped with an online carbonate device (Kiel IV) at the Institute of
Geology and Geophysics, Chinese Academy of Sciences and Isotope Laboratory of Xi'an Jiaotong University. The
powdered subsamples weighing ~30 μg were drilled along the central growth axis using a Micromill device and then
reacted with ~103 % phosphoric acid at 70 °C. The stable oxygen and carbon isotopic compositions of the generated
$CO_2$ gas were measured with working $CO_2$ standard gas whose values have been calibrated by international standards.
All results are reported as the per mil deviation relative to the Vienna Pee Dee Belemnite (VPDB). The reported
precision of both $\delta^{18}O$ and $\delta^{13}C$ at 1σ level is better than 0.1 ‰.
Trace element ratios (Mg/Ca, Sr/Ca, Ba/Ca), of which the intensity of emission lines are 285.2 nm, 407.8 nm, and
373.7 nm, were measured using Laser Induced Breakdown Spectroscopy (LIBS) following the detailed description in
Li et al. (2018). In brief, analyses were performed by pulsing laser and then analyzing the intensity of specific spectrum
of trace elements to obtain their content and ratios relative to the calcium for each point. The obtained record is the
median intensity ratio based on 20 pulses at each sampling site after 5 laser shots for pre-cleaning the surface. The
measurements were performed continuously along the speleothem's growth axis at 0.3 mm increment and a total of
565 data were obtained.
**3 Results**
**3.1 $^{230}$Th dates and age model**
The $^{230}$Th dating results of the BH-2 are presented in Table S1 which shows that the BH-2 covers the interval between
9.0 and 7.9 ka BP. All dates are in stratigraphic order within uncertainties. The average dating uncertainty is ± 57
years at 2σ level. For the period from 8.25 to 8.11 ka BP, we present the speleothem record from Duan P et al. (2023),
which is based on the combination of the annual lamina counting and $^{230}$Th dates. In addition, here we use an updated
chronology of the BH-2 based on the Oxcal algorithm (Ramsey, 2008), which includes ten additional ages from the
remnant sections (Figure 2b).
**3.2 Stable isotopic compositions and growth rate**
The BH-2 record contains 663 pairs of $\delta^{18}O$ and $\delta^{13}C$ data with a mean temporal resolution of ~1.6 years. The $\delta^{18}O$
values range from -7.1 ‰ to -11.5‰ with a mean of -9.3 ‰ and $\delta^{13}C$ values vary from -8.0 ‰ to -12.1 ‰ with an
average value of -10.2 ‰ (Figures 2d and 2e). It can be seen that the $\delta^{13}C$ profile follows the same general patterns as
the $\delta^{18}O$ ($r = 0.63$, $p < 0.01$). Compared to the later stage, although some fluctuations are included, the $\delta^{13}C$ and $\delta^{18}O$
profiles are relatively invariable before 8.5 ka BP. In contrast, the $\delta^{18}O$ record exhibits a remarkable positive shift at
~8.45–8.39 ka BP, during which period the $\delta^{13}C$ record shifts less prominently to the positive direction but with a
fluctuating increasing trend. The rebound from the positive shift of $\delta^{13}C$ and $\delta^{18}O$ profiles is followed by a less variable
episode spanning 8.39–8.26 ka BP. Afterward, as the most remarkable feature, both records show extremely positive
excursions spanning ~8.26–8.14 ka BP (Figure 2). The positive anomaly is followed by a shift to the opposite extreme





to reach the most negative stage in the $\delta^{18}O$ record during 8.14–8.05 ka BP, which is not conspicuous in the $\delta^{13}C$
record.
The growth rate of the BH-2 was established based on the reconstructed chronology (Figure 2c). It is apparent that
speleothem BH-2 was contiguously deposited without visible growth hiatus and the growth rate during 8.46–8.16 ka
BP (< 0.15 mm/year) is apparently lower relative to other intervals (> 0.15 mm/year on average). Specifically, there
are obvious transitions from higher to lower growth rates at ~8.46 ka BP and in the opposite trend at ~8.16 ka BP.
Moreover, it is notable that the lowest growth rate from ~8.28 to 8.18 ka BP broadly corresponds to the relatively high
$\delta^{18}O$ and $\delta^{13}C$ excursions.
**3.3 Trace element ratios**
The signals in the trace element ratio records are quite variable (Figure S2). Similar to $\delta^{18}O$ and $\delta^{13}C$ records, all of
the Mg/Ca, Sr/Ca, and Ba/Ca records display positive excursions at ~8.40 and 8.20 ka BP despite the relative
ambiguity of the former one in the Sr/Ca and the latter one in the Mg/Ca, respectively. Besides, there is another more
positive excursion at ~ 8.86 in the Mg/Ca ratio record, which is absent in the other two records. After principal
component analysis of the three records, the excursions at ~8.40 and 8.20 ka BP are especially conspicuous (Figure
2f). Before 8.46 ka BP, the PC1 result fluctuates frequently with considerable magnitude, which seems coincident
with the $\delta^{18}O$ variability. In the duration of 8.46–8.38 ka BP, it exhibits a fluctuating positive trend and a rapid rebound
at ~8.38 ka BP. Aftermath, the values remain relatively stable until ~8.23 ka BP when another positive excursion
commences. In this excursion, the PC1 values culminate at ~8.12 ka BP followed by a rapid rebound which indicates
the termination of this excursion. The values remain relatively stable after 8.10 ka BP.
**4 Discussion**
**4.1 Proxy interpretations**
The replication test of $\delta^{18}O$ records between the BH-2 from Huangyuan Cave and the KLS12 from nearby Kulishu
Cave (Duan W et al., 2021) by using the ISCAM (Intra-site Correlation Age Modeling) algorithm (Fohlmeister, 2012)
show significantly positive correlation ($r = 0.62$, $p < 0.05$) during 9.0–7.9 ka BP (Figure S3), strongly suggesting that
the influence of kinetic fractionation is likely insignificant and the carbonate deposition process is close to equilibrium
(Dorale and Liu, 2009). Hence, the BH-2 $\delta^{18}O$ signals reflect the changes in drip water $\delta^{18}O$ which in turn inherit from
$\delta^{18}O_p$ related to the regional hydroclimate variations in general. Notably, the study site is located along the summer
monsoon fringe with relatively low annual precipitation, and thus the thermodynamics variations in EASM in the
areas can significantly bias the mean annual $\delta^{18}O$ value, e.g., the summer rainfall amount. Indeed, the modern
observations (Duan al., 2016) and reanalysis results (He et al., 2021; Duan P et al., 2023; Zhao et al., 2023) have
proved that speleothem $\delta^{18}O$ in the study area can be used as a reliable proxy to indicate the regional precipitation
variations and the dynamic changes of the summer monsoon circulation, that is, depleted $^{18}O$ corresponds to increased
rainfall over the study area and strengthened EASM, and vice versa.



Under the equilibrium fractionation conditions, the carbon isotope ratios ($\delta^{13}C$) of speleothem carbonate reflect a
mixture of three carbon sources: plant root-respired $CO_2$ in the soil zone, atmospheric $CO_2$, and dissolution of bedrock
carbonate (McDermott, 2004), in which the plant-related $CO_2$ is the most important for the variability of the
speleothem $\delta^{13}C$ (Fairchild et al., 2006; Li Y et al., 2020). It has been suggested that changes in the density of
vegetative cover and biomass exert a critical impact on the speleothem $\delta^{13}C$ variations in the study region, instead of
the relative ratio of C3 (woody taxa) and C4 (grasses) plants (Duan et al., 2014). This is consistent with our observation
that the $\delta^{13}C$ values of speleothem BH-2 fall between -8 and -12 ‰, which is within the typical range for the C3-
dominant plant coverage (McDermott, 2004; Fairchild et al., 2006). Although climate-induced changes in the karst
system, like $pCO_2$ degassing, water infiltration, and prior calcite precipitation (PCP) could also contribute to the $\delta^{13}C$
changes (Fairchild and Treble, 2009; Li et al., 2020), the significant covariance of $\delta^{13}C$ and $\delta^{18}O$ in the BH-2 and
minor effect of kinetic fractionations as aforementioned, as well as the unbiased $\delta^{18}O$ signal inherited from
precipitation strongly suggest that the density of vegetative cover, the biomass activity, and the vadose of seepage
solution dominated by regional hydroclimatic conditions could play a crucial role in the decadal to centennial scale
variations of $\delta^{13}C$ in speleothem BH-2.
The influence of PCP can be inferred from trace element concentrations such that strong (weak) PCP normally induces
a high (low) trace element content relative to the calcium in the speleothem calcite (Johnson et al., 2006; Fairchild
and Treble, 2009). In general, higher trace element ratio values indicate overall drier conditions when reduced
infiltration and increased residence time in the epikarst above the cave favors faster $CO_2$ degassing and PCP, inducing
relatively higher trace element content in the cave drip-water due to the preferential loss of $Ca^{2+}$ along the deposition
path; the opposite processes occur in wetter conditions (e.g., Cruz et al., 2007; Griffiths et al., 2010; Zhang et al.,
2018). On the other hand, water-rock interaction may have been enhanced in the aquifer during direr conditions
because of the prolonged residence time of fluid in the path way, which tends to favor the leaching of Mg and Sr
element from the dolomite host rock (Fairchild et al., 2000) and eventually leads the two elements to enrichment in
dripwater, and hence speleothem. Apparently, both above two mechanisms indicate the trace element ratios can be
used as a reliable proxy of local wetness conditions. Regarding the speleothem growth rate, the sharp drops and
persistent lower values in this proxy corresponding to major positive $\delta^{18}O$ and $\delta^{13}C$ excursions signify that it most
likely was controlled by a sufficient or insufficient supply of drip water, and hence the local rainfall amount (e.g.,
Polyak et al., 2004; Banner et al., 2007).
In summary, the broad similarity of multi-proxies ($\delta^{18}O$, $\delta^{13}C$, trace element ratios, and growth rate) in speleothem
BH-2 lends robust support to that all of them record changes in hydroclimatic characteristics (Fairchild and Treble,
2009), that is, the intensity of the EASM and associated rainfall amount presumably dominating the hydroclimatic
variabilities over and in the cave in the study area. On the other hand, the discrepancy between various proxies could
suggest that different factors exert influence on these signals in the meteoric water-cave aquifer-drip water-carbonate
precipitation processes.



**4.2 Climate fluctuations between 9.0 and 7.9 ka BP in Beijing**

The variability of the BH-2 $\delta^{18}$O record reveals inter-decadal to multi-decadal dry (> +1$\sigma$) or pluvial (< -1$\sigma$) oscillations from 9.0 to 7.9 ka BP without a distinct long-term trend (Figure 2). One noticeable feature of our $\delta^{18}$O record is a switch from relatively muted to highly variable episodes divided at ~8.5 ka BP, consistent with the absence and dominance of centennial to inter-decadal periodicity before and after 8.5 ka BP, respectively (Figure 2).

The first persistent drought, indicated by positive $\delta^{18}$O excursion exceeding +1$\sigma$ values for more than 15 years, initially started at 8.46 ka BP and terminated at 8.39 ka BP (8.4 ka event herein). The entire event is characterized by a saw-tooth structure with a dramatic 2.5 ‰ increase within ~55 years and a 2.2 ‰ rebound within 11 years, indicating a fast weakened EASM and thus reduced precipitation in the study area. This arid condition is supported by the contemporaneous trace element records which show a remarkable positive shift that seems strictly resemble the $\delta^{18}$O record regarding both the shape and duration, pointing to the changed dynamic process in the cave in response to the decreased precipitation water supply. Additionally, the high-to-low transition of growth rate commencing ~8.46 ka BP presumably results from less drip water supply and further in turn reduced precipitation over the cave, marking the start of the EASM weakening. However, the change of vegetation indicated by the $\delta^{13}$C proxy is not immediate. It seems that the increasing $\delta^{13}$C trend begins later than other proxies and only exhibits a short excursion, probably indicating the nonlinear response of vegetation evolution to the hydroclimate change, especially in a short-time climate event. This could be related to the delayed shortage of subground water for plant growth and a muted response of ecological processes to the hydroclimatic variability in a relatively wet context as indicated by low $\delta^{18}$O and trace element values surrounding this excursion (Duan P et al., 2021).

Following the end of above arid excursion, another centennial oscillation in much temperate mode persisted to ~8.25 ka BP. Subsequently, the BH-2 $\delta^{18}$O exhibited the most remarkable droughts with centennial positive excursion between ~8.26 and 8.11 ka BP, conservatively corresponding to the 8.2 ka event (Duan P et al., 2023). This drought event is also proved in the trace element records via the increased values, in concert with the decreased seepage water and hence enhanced PCP. In detailed structure, these trace element ratio records commonly show prominent positive excursion at ~8.20 and 8.14 ka BP, the latter of which is especially elevated in them. However, the slowly increased pattern in the trace element ratio records from 8.26 to 8.18 ka BP is quite distinct from the $\delta^{18}$O record in which its values dramatically increase in the first 70 years, suggesting the probably nonlinear relationship between regional climate ($\delta^{18}$O) and local hydroclimatic condition (trace element ratios). Moreover, in this event, the $\delta^{13}$C exhibits a prominent positive shift, pointing to the decay of the ecosystem in this severe drought event. It is noteworthy that the variation pattern of $\delta^{13}$C in the 8.2 ka event is more similar to the $\delta^{18}$O relative to the 8.4 ka event. This absence of muted $\delta^{13}$C signal suggests the close relationship between the vegetation and regional hydroclimatic conditions in a long duration and more severe climatic deterioration. Intrigueingly, the lower excursion of growth rate somehow predates other proxies. This inter-proxy discrepancy suggests that there are other potential factors, such as the temperature (Wong et al., 2015), controlling the cave dynamic processes, and the growth rate could be a more qualitative indicator to broadly constrain the hydroclimatic conditions in combination with other proxies.

Afterward, the hydroclimatic conditions go to the reverse side of the extreme, manifesting a multi-decadal excessive rebound (i.e., overshoot) attaining the lowest $\delta^{18}$O values (-11.5 ‰) of the entire record, suggesting the strongest



pluvial event (Duan P et al., 2023). This overshoot is additionally supported by trace element ratio record which show
quite low values relative to the period before 8.46 ka BP. However, the rebound of the $\delta^{13}$C during the post-8.2 ka
event is not as conspicuous as the $\delta^{18}$O overshoot and only reaches the mean level of that preceding the 8.4 ka event.
These features further illustrate the aforementioned nonlinear relationship among the variabilities of regional climate,
local hydrological condition, and ecosystem. In other words, the coverage of vegetation and soil microbiological
activity during the overshoot event didn't recover to the initial conditions before the 8.2 ka event.
The different behavior of $\delta^{13}$C after two similar severe droughts at 8.40 and 8.20 ka BP suggests the degree of resilient
ecosystem to the different rebound rainfall intensity. For the 8.40 ka event, the subsequent rebound of $\delta^{13}$C to its prior
value suggests the high-level resilience of the plant community to environmental variations under the moderate
precipitation amount as indicated by the $\delta^{18}$O and trace element ratio records. In contrast, the suddenly excessive
increase of precipitation after the 8.2 ka event, which was much more than that before the event, could have suppressed
the recovery of vegetation and soil biological activity and thus the moderate rebound of $\delta^{13}$C values. Theoretically,
the longer weakened atmospheric circulation during the 8.2 ka event and reduced precipitation presumably induced
deteriorated vegetation as well as poor-developed soil. However, it seems that the precipitation intensity after the 8.2
ka event exerted a key role on the recovery of vegetation density and soil productivity. Specifically, the severe 8.2 ka
drought event had a profoundly negative impact on the vegetation-soil system and led them to become more vulnerable
under the water shortage conditions. On the other hand, the excessive precipitation after this drought could cause soil
erosion and further ecological damage, suppressing the ecosystem recovery above the cave as well as the $\delta^{13}$C signals
in speleothem. Conclusively, the ecosystem in this karst region was quite vulnerable and the variability of the
vegetation-soil system here was tied to local hydrologic conditions with both high and low thresholds.
To summarize, akin to the $\delta^{18}$O record, other proxy records of the BH-2 (Figure 2) delineate two major drought events,
indicated by prominent excursions centered at 8.40 and 8.20 ka BP, respectively, suggesting vegetation degeneration
(Duan et al., 2014) and elevated prior calcite precipitation (PCP) arising from longer residence time of solution in the
karst aquifer (e.g., Johnson et al., 2006; Fairchild et al., 2009), both of which responded to the deteriorated
hydroclimatic conditions. The discrepancy between them could suggest that other drivers than only hydroclimatic
conditions possibly have played a non-negligible role in the processes of speleothem formation. In particular, the
intensity of the EASM ($\delta^{18}$O) and the precipitation amount (trace element ratio) over the study area presumably were
definitely correlated on a broad pattern but did not necessarily exactly follow each other.
**4.3 Spatial patterns for the two drought-one pluvial pattern and underlying mechanisms**
This two drought-one pluvial pattern from 8.5 to 8.0 ka BP in speleothem BH-2 represents global scale climate
disturbance signals rather than a regional phenomenon since these climate excursions have been widely documented
(Figures 3 and 4). In the ASM domain, speleothem records from such as Lianhua (Dong et al., 2018), Wuya (Tan et
al., 2020) Caves in North and Northwest China, and Qingtian Cave (Liu et al., 2015) in central China exhibit consistent
structure with the BH-2 at around 8.2 ka BP. In particular, a broad anomaly spanning ~340 years between 8.46 and
8.12 ka BP has been revealed (Tan et al., 2020) and we find the post-8.2 ka overshoot is also distinguishable (Figure
4) in the speleothem $\delta^{18}$O record from the western Chinese Loess Plateau which is situated in the northern limit of the



ASM. Unlike these north-located records, although a prominent 8.2 ka event is documented in speleothem of Heshang
Cave in central China (Liu et al., 2013), the preceded excursion is ambiguous and the post-8.2 ka event anomaly is
absent. Coincidentally, a similar phenomenon seems to occur in Dongge Cave in South China (Cheng et al., 2009). This
probably suggests that relative to the low latitudes, the climate in the north part of the ASM is more sensitive to the
climate perturbation signals originating from the high northern latitude regions because high northern latitude climate
variations can strongly affect the westerly changes and finally influence the EASM (Chiang et al., 2015; Duan et al.,
2016; Tan et al., 2020). In the low latitudes of the Indian summer monsoon realm, the speleothem δ¹⁸O record from
Hoti Cave is remarkably consistent with the pattern in our record. Specifically, Hoti Cave record shows positive δ¹⁸O
excursions by ~2 ‰ in amplitude centering ~8.4 ka BP and a growth hiatus at 8.2 ka BP surrounded by enriched ¹⁸O,
pointing to the drought conditions due to the weakened Indian summer monsoon attendant with a southward shift of
the intertropical convergence zone (ITCZ). After the growth resumption, an overshoot can be identified (Cheng et al.,
2009). It happens that the two positive excursions are quite pronounced in nearby Qunf Cave (Figure 3) (Cheng et al.,
2009), whereas the overshoot is absent. Collectively, records from more sensitive areas in the ASM domain intactly
preserved the two drought-one pluvial pattern, while the pre-8.2 ka event or the overshoot is missed in records from
insensitive regions.
In the North Atlantic region, Greenland ice core δ¹⁸O (Thomas et al., 2007) and reconstructed temperature based on
argon and nitrogen isotopes (Kobashi et al., 2017) captured both the 8.2 ka event and ensuing overshoot, and the pre-
8.2 ka event is apparent in the temperature profile but ambiguous or slightly excursed (Jennings et al., 2015) in the
δ¹⁸O records. Indeed, the atmospheric circulation over Greenland has substantially changed since ~8.5 ka BP as
suggested by increased potassium and calcium ions, indicators of dust supply to Greenland, as well as decreased snow-
accumulation rate (Rohling and Pälike, 2005; Kobashi et al., 2017; Burstyn et al., 2019). The absent signal of the pre-
8.2 ka event in δ¹⁸O records could be attributed to the compensation of other processes like precipitation seasonality
and summer warming (He et al., 2021). The prolonged climate anomalies around 8.2 ka BP are further supported by
two negative anomalies at 8.3 and 8.2 ka BP, respectively, in northern Spain speleothem δ¹⁸O record (Domínguez-
Villar et al., 2009), lower tree ring width from 8.42 to 8.0 ka BP in Germany (Spurk et al., 2002), as well as degraded
climate conditions between 8.45 and 8.10 ka BP revealed by speleothem proxies from Père Noël Cave in Belgium
(Allan et al., 2017). All of these collectively suggest a series of pronounced climate oscillations between 8.5 and 8.0
ka BP, instead of merely the 8.2 ka event, is of hemispheric significance (Rohling and Pälike, 2005).
Similar but antiphase patterns are observed in the records from the Southern Hemisphere. For example, it appears that
speleothem record from Lapa Grand Cave in East Brazil (Stríkis et al., 2011) captured the two pluvial-one drought
structure (Figures 3 and 4). Intriguingly, speleothem record from Padre Cave (Cheng et al., 2009) fails to preserve as
clear pre- and post- 8.2 ka events as its adjacent Lapa Grand Cave (Figure 4), presumably due to different cave settings.
But, the beginning deposit of speleothem in Padre Cave at ~8.5 ka BP, coeval with the reduced precipitation in the
ASM domain, likely reflects more favorable hydroclimatic conditions due to more precipitation, which in turn could
arise from intensified South American summer monsoon associated with the southward displacement of the ITCZ
(Wang X et al., 2004), suggesting the possible occurrence of the pre-8.2 ka event there. Coincidentally, the speleothem
growth resumption after a long hiatus (Duan P et al., 2021), together with the negative trend of speleothem δ¹⁸O record





(Voarintsoa et al., 2019) in Northwest Madagascar commenced at ~8.5 ka BP and persisted until the end of the 8.2 ka
event, indicative of more precipitation in response to the southward ITCZ shift, suggesting the extent of the pre-8.2
ka event to the East Africa monsoon domain. However, the post-8.2 ka event was not clearly identified by the
Northwest Madagascar record and thus more evidence is needed.
The two droughts-one pluvial pattern revealed in our BH-2 records could mainly correspond to the waxing and waning
of drainages of the LAO (Barber et al., 1999; Ellison et al., 2006) and contemporary ice sheet melted freshwater flux
(Matero et al., 2017, 2020) (Figure 3), both of which causally related to the AMOC strength dynamics. Firstly, the
major two-step outburst of the LAO (e.g., Ellison et al., 2006; Kleiven et al., 2008; Jennings et al., 2015; Lochte et al.,
2018; Godbout et al., 2019, 2020) and the continuous Laurentide Ice Sheet (LIS) melting together contributed to the
increase of total freshwater flux (e.g., Morrill et al., 2014; Matero et al., 2017, 2020), inducing observed sea level rise
in North Atlantic commencing ~8.5 ka BP (Hijma et al., 2010), cooling conditions initially in the circum-North
Atlantic region and perturbed into other areas through fast atmospheric propagations (Cheng et al., 2009, 2020; Liu et
al., 2013; Buizert et al., 2014; Duan P et al., 2021). Coincident with enriched $^{18}O_p$ in most ASM domains, the intensity
of the East Asian summer monsoon was weakened (Cheng et al., 2009) and less precipitation fell in the Beijing area
(Duan P et al., 2023). In contrast, due to the southward displacement of the ITCZ in response to the hemispheric
thermal contrast, the Southern Hemisphere, like Northeast Madagascar and East Brazil, received more precipitation
(i.e., stronger monsoon) and thus speleothem records there exhibit depleted $^{18}O_p$. Further, the simulated smaller
freshwater flux peak at ~8.5 ka BP relative to the second one at 8.2 ka (Figure 3) (Matero et al., 2020) could provide
a potential explanation for the lower amplitude and shorter duration of the pre-8.2 ka event relative to the 8.2 ka event
in our record and the absence of the pre-8.2 ka event in other records. Additionally, the 8.2 ka event is preceded by a
remarkable reduction in solar activity by ~1 Wm$^{-2}$ with a duration of ~150 years, beginning at ~8.45 ka BP (Rohling
and Pälike, 2005; Steinhilber et al., 2009; Wanner et al., 2011; Burstyn et al., 2019), and an increase in the magnitude
and frequency of volcanic eruptions (Kobashi et al., 2017; Burstyn et al., 2019), both of which are also thought to
contribute to the prolonged climate disturbance via different impacts on atmospheric processes.
On the other hand, the overshoot in the ASM domain could be remotely related to the higher temperature in the North
Atlantic (Kobashi et al., 2017; Andersen et al., 2017) (Figure 4) which in turn possibly arose from the remarkably
speed-up AMOC (Ellison et al., 2006; Renold et al., 2010; Mjell et al., 2015; Andersen et al., 2017). The accelerated
AMOC led to more heat release in the North Atlantic and anomalously strengthened ASM. In the meanwhile, the
ITCZ and associated rainbelt were displaced northwards, causing less precipitation in east Brazil as evidenced by
positive δ$^{18}$O excursion of speleothem from Lapa Grande Cave (Figure 4).
**5 Conclusions**
The multi-proxy records of speleothem BH-2 document the multi-decadal to centennial scale hydroclimate changes
in Beijing of North China with two arid episodes at ~8.4 and 8.2 ka BP, and an immediately ensuing excessive rebound
after the 8.2 ka event. A comparison with other paleoclimate records suggests that these prominent climate fluctuations
with two drought-one pluvial pattern should be a global signal instead of a regional phenomenon. We propose that the
slowdown and resumption of the AMOC controlled by the freshwater flux into the North Atlantic and the resultant



reorganization of the atmospheric circulation during the study stage mainly contribute to the arid and pluvial
excursions, and the influence of volcanic outbursts and reduced solar activity are also non-negligible.
**Data availability**
All data needed to evaluate the conclusions in the paper are presented in the paper. The data will be archived at the
NOAA National Climate Data Center (https://www.ncdc.noaa.gov/data-access/paleoclimatology-data) when this
manuscript is accepted.
**Author contributions**
PD, HL and HC designed the research and experiments. PD wrote the first draft of the paper. HL, HC, and AS
revised the paper. ZM did the fieldwork and collected the samples. ZM and HC conducted the $^{230}$Th dating. ZM,
HC, and PD conducted the oxygen isotope measurements. All authors discussed the results and provided inputs on
the paper.
**Competing interests**
The authors declare that they have no conflict of interest.
**Acknowledgments**
This work was supported by the National Natural Science Foundation of China grants (42150710534 and 41888101
to H.C.). We specially thank Ming Tan and Wuhui Duan from Institute of Geology and Geophysics, Chinese Academy
of Sciences for their helpful suggestions.

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

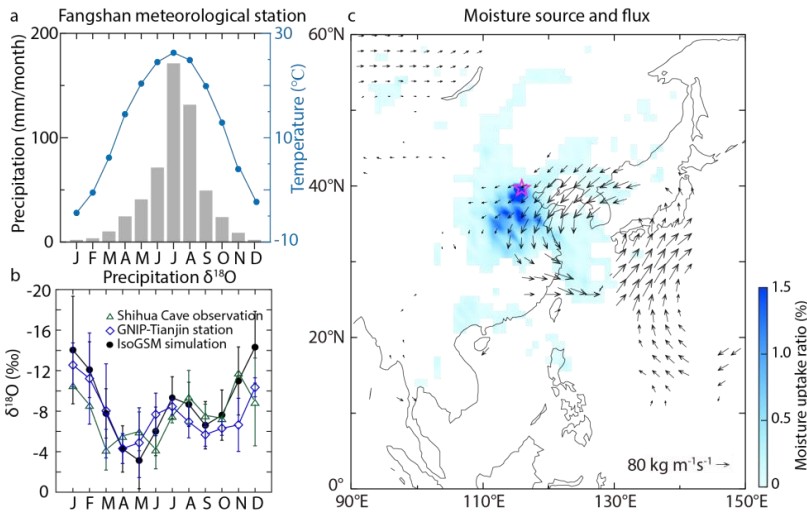

**Figure 1. Climatology and locations. (a)** Climographs of precipitation amount (gray bars) and temperature (blue dots connected with lines) at Fangshan Station (39°46′N, 116°28′E) near the study site, based on Chinese Meteorological Administration data (http://www.cma.gov.cn/). **(b)** Annual cycle comparison of $\delta^{18}O_p$ from observations of GNIP Tianjin station (https://www.iaea.org/services/networks/gnip) (1988–2002 with absent data covering 1993–2000, blue triangles), Shihua Cave (Duan et al., 2016) (2011–2014, green diamonds), and IsoGSM-simulation data (Yoshimura et al., 2008) (1979–2017, black dots) at the Huangyuan Cave. Error bars represent the 1σ uncertainty of $\delta^{18}O_p$ values for each month. **(c)** Mean July-August (JA) moisture source region (blue shading) the Hybrid Single Particle Lagrangian Integrated Trajectory (HYSPLIT) model version 4.0 (Stein et al., 2015) based on the NOAA-NCEP/NCAR reanalysis global meteorological field data of 2010–2020 (Sodemann et al., 2008; Krklec and Dominguez-Villar, 2014) and water vapor flux (arrow) from the European Centre for Medium-Range Weather Forecasts Reanalysis fifth-generation dataset (ERA5) (Hersbach et al., 2020) between 1980 and 2015.

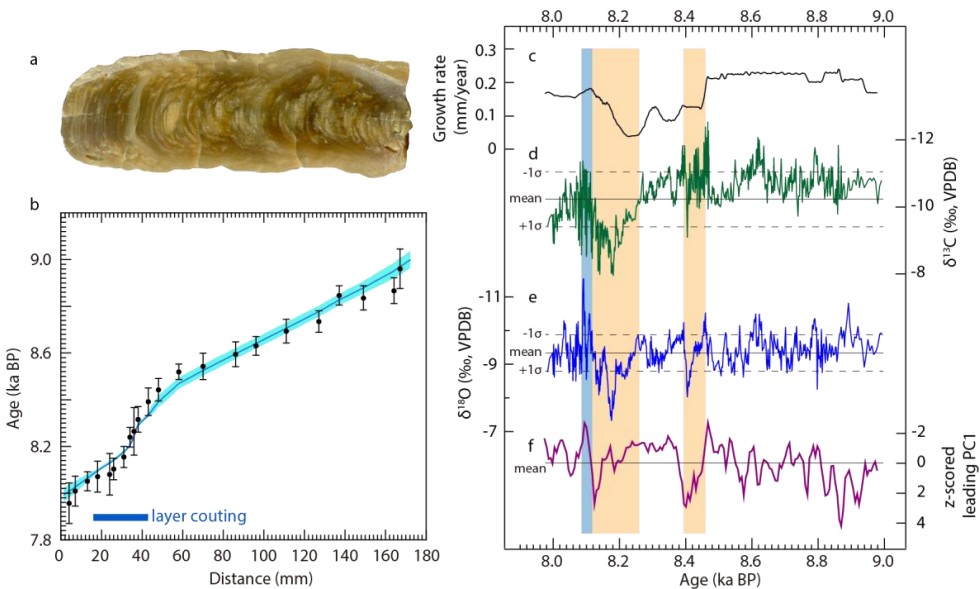

**Figure 2. Age model and proxy profiles of speleothem BH-2.** (**a**) Scanned image of speleothem BH-2. (**b**) Oxcal-derived age model (blue, Ramsey, 2008) with 95 % confidence interval (light blue shading). Black error bars on $^{230}$Th dates represent 2σ analytical errors. The horizontal blue bar marks the range with layer counting. (**c**) The inferred growth rate of the BH-2 based on the chronology in (**b**). (**d**) and (**e**) are δ$^{18}$O (dark blue) and δ$^{13}$C (green) profiles, respectively. The mean (solid) and the ±1σ values (dashed) for each entire record are indicated by the horizontal lines. (**f**) 30-year loess filtered *z*-scored leading PC record of trace element ratios of Ba/Ca, Mg/Ca, and Sr/Ca (see Figure S2). The mean value of the PC1 record is presented. The vertical yellow bars in the right subpanel mark the anomalously positive episodes and the light blue bar indicates the subsequent δ$^{18}$O overshoot after the 8.2 ka event.





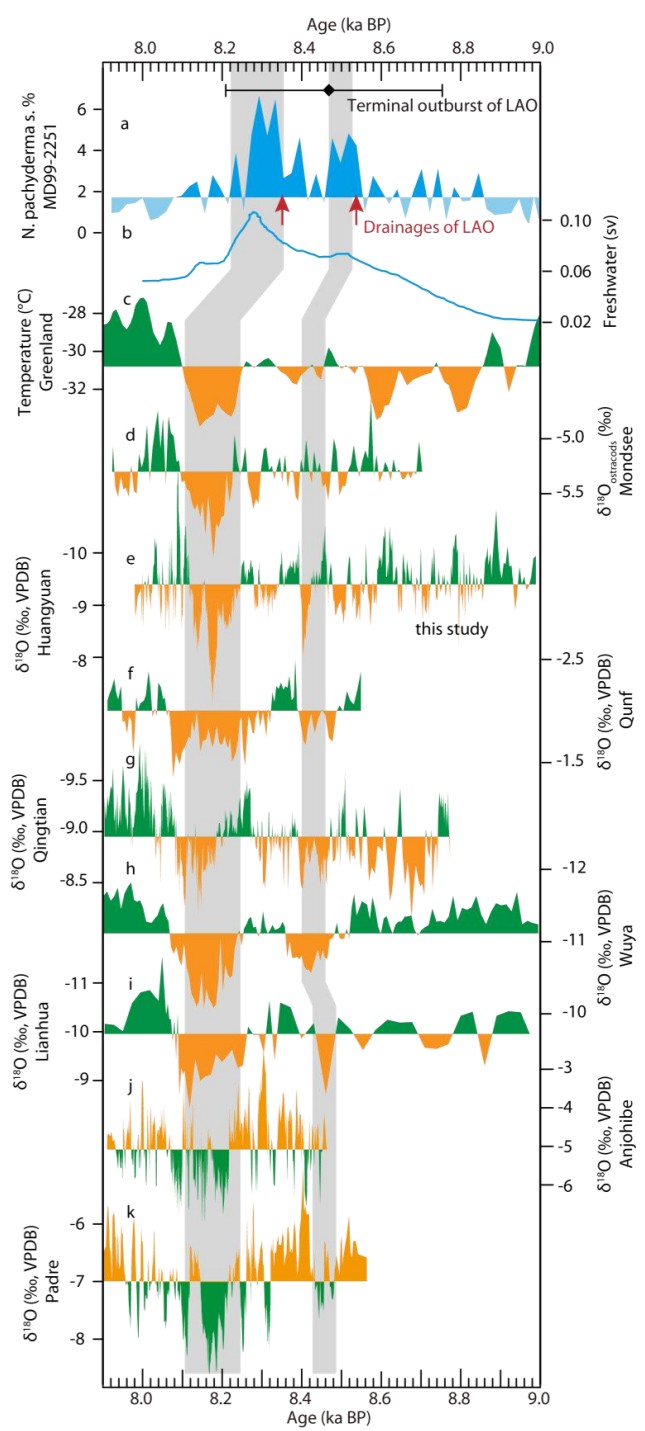



**Figure 3. Comparisons of the BH-2 δ¹⁸O record with records from circum-North Atlantic, ASM domain and South America.** (**a**) *N.pachydermas.abundance* record from MD03-2665, North Atlantic (Ellison et al., 2006). The black diamond and error bar on the top indicate the dating of terminal outburst of LAO (Barber et al., 1999). The red arrows point to the two-step drainages of LAO into the North Atlantic. (**b**) Modelled freshwater flux from Laurentide Ice Sheet in unit of Sverdrups (Sv) (Matero et al., 2020). (**c**) Reconstructed temperature in Greenland (Kobashi et al., 2017). (**d**) δ¹⁸O$_{ostracods}$ record from Modesee, Austria (Andersen et al., 2017). (**e**) The BH-2 δ¹⁸O record from Huangyuan Cave, Beijing (this study). (**f**) High-resolution δ¹⁸O record (Fleitmann et al., 2003) from Qunf Cave with more precise ages (Cheng et al., 2009) (**g**) δ¹⁸O record from Qingtian Cave, China (Liu et al., 2015). (**h**) δ¹⁸O record from Wuya Cave, Northwest China (Tan et al., 2020). (**i**) δ¹⁸O record from Lianhua Cave, North China (Dong et al., 2018). (**j**) δ¹⁸O record from Anjohibe Cave, Northwest Madagascar (Duan P et al., 2021). (**k**) High-resolution δ¹⁸O record from Padre Cave, Brazil. The δ¹⁸O scale of **j–k** is inverse to other speleothem records. The vertical gray shading bars indicate the events centered at 8.4 and 8.2 ka BP.



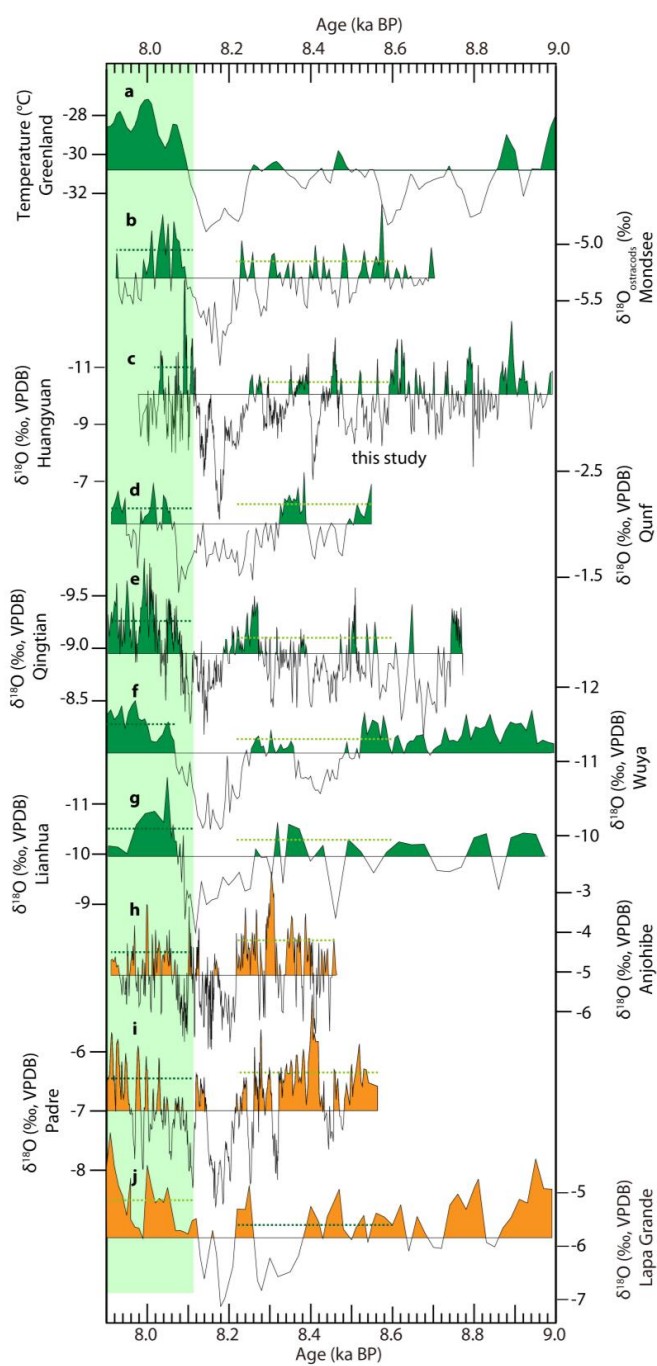

**Figure 4. Comparisons of the overshoot between the BH-2 δ¹⁸O record with other records. (a)** Reconstructed temperature in Greenland (Kobashi et al., 2017). **(b)** $\delta^{18}O_{ostracods}$ record from Modesee, Austria (Andersen et al., 2017). **(c)** The BH-2 δ¹⁸O record from Huangyuan Cave, North China (this study). **(d)** High-resolution δ¹⁸O record

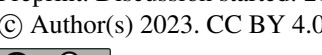



from Qunf Cave, Oman (Fleitmann et al., 2003: Cheng et al., 2009) based on more precise [230]Th dates (Cheng et al., 2009). (**e**) $\delta^{18}$O record from Qingtian Cave, Central China (Liu et al., 2015). (**f**) $\delta^{18}$O record from Wuya Cave, Northwest China (Tan et al., 2020). (**g**) $\delta^{18}$O record from Lianhua Cave, North China (Dong et al., 2018). (**h**) $\delta^{18}$O record from Anjohibe Cave, Northwest Madagascar (Duan P et al., 2021). (**i**) High-resolution $\delta^{18}$O record from Padre Cave, Brazil, on the Oxcal-derived chronology based on the [230]Th dates of Cheng et al. (2009). (**j**) $\delta^{18}$O record from Lapa Grande Cave (Stríkis et al., 2011) in Brazil. The $\delta^{18}$O scale of **h**–**j** is inverse to other speleothem records. The vertical green shading bar represents the overshoot episode following the 8.2 ka event. The $\delta^{18}$O value lower than the mean value of the entire records from the Northern Hemisphere, and Greenland reconstructed temperature record higher than the mean value of the entire record is shaded in green. The $\delta^{18}$O values higher than the mean value of the entire records from the Southern Hemisphere are shaded in brown. The green horizontal dashed lines in each record indicate the mean $\delta^{18}$O values for the age range they cover before (8.60–8.22 ka BP) and after (8.10–7.90 ka BP) the 8.2 ka event.