# Peer review of "A series of climate oscillations around 8.2 ka BP revealed through"

_Climate of the Past, 2023_

## Community Comment (CC1)

Duan et al. provide a new high resolution 8.2ka speleothem record spanning 9.0-7.9ka BP period in north China.They reconstructed the time series over this period using a variety of indicators, including carbon and oxygen isotopes, ratios of trace elements, and growth rate.The authors identified two drought periods of 8.4 and 8.2ka and found that the behavior of carbon and oxygen isotopes and trace elements was different, which may be responsible for the nonlinear response of the local ecosystem.They suggest that there were several centennial scale climate fluctuations around the 8.2ka event, and that two droughts-one pluvial pattern between 8.5-8.0ka were a widespread event on a global scale and were closely related to the northern high latitude.The manuscript is carefully prepared, this record is relatively rare in northern China, and the age model is accurate, the resolution is super high,after a certain amount of thought and revision this paper is recommended for publication.

**General notes**

1. Cheng et al.(2009) and your last paper( Duan et al.,2023) reported that the 8.2ka event had a two-stage structure,but this manuscript and Tan et al.(2020) believe that there is a two drought-one pluvial pattern at 8.2ka and 8.4ka, does this similar expression give rise to some misunderstanding? Because in some papers 8.4ka and 8.2ka together constitute the 8.2ka cold event.

2. What are the periodic changes of trace elements, oxygen isotopes and carbon isotopes? Do they have a common period and are they influenced by Solar Output, AMO, PDO or even ENSO on a short time-scale?

3. Can the $\delta^{234}U_{initial}$ attached to the dating results be used as an indicator to reflect these climatic events?Can the carbon isotope at 8.4ka be considered a wet event?

4. Carbon isotopes vary much more than oxygen isotopes, as is the case in Hulu cave. It is generally assumed that carbon isotopes are more sensitive to climate change.Why is there a nonlinear response now?

**Specific notes**

1. Figure2a, this paper is based on Duan et al.,2023 (GRL) to add new data for research. Please distinguish the ages used in the previous paper and the new ages added in this paper in this figure.Figure2f, from 9.0-8.5ka, There is a clear decreasing trend in the PC1 index of trace elements in this period, indicating that the climate is becoming wetter, which is not reflected in oxygen and carbon isotopes.

2. Figure3, in addition to the dating error of LAO you have marked, the marking of the dating error bar of other stalagmites records is conducive to the understanding of this paper.

3. Figure s2, what are the correlation between the three trace elements, if there are calculations of the correlation, you can show their consistency.

---

## Author Comment (AC1)

This manuscript demonstrates a series of high-resolved multi-proxy records between 9.0 and 7.9 ka BP based on a stalagmite named BH-2 with precise chronology from North China. The authors try to show a complete picture of the pre-, at and post-event of the 8.2 ka at centennial time scales, which did not receive enough concern before. However, the manuscript needs to be further improved since the advantages of the multi-proxy records have not been fully utilized.

Though the authors demonstrate the records of d$^{13}$C and PC1 of trace element ratios besides d$^{18}$O during the same time, the d$^{18}$O is still the only proxy to be used to interpret the climate oscillations around the 8.2 ka BP. As d$^{18}$O is a climate signal mixed with local and circulation information (i.e., EASM), whereas other proxies, like d$^{13}$C and PC1 of trace element ratios mainly reflect the local climate change, the authors actually could try to find a way to separate the local and remote information in BH-2 d$^{18}$O base on other proxies to improve the interpretation of d$^{18}$O.

Re: We thank the reviewer's concern about the issue of proxies' interpretation. We have to admit that it is difficult to find a way to separate the local and remote information in BH-2 $\delta^{18}$O based on other two proxies, and this issue is a little beyond the scope of this study. Actually, the interpretation of speleothem $\delta^{18}$O has long been discussed for decades of years and it is not easy to detangle the detailed inner processes because this proxy can be affected by many factors. It is now a consensus that speleothem $\delta^{18}$O in ESAM region could indicate large scale consistent atmospheric circulations, which is supported by our previous reanalysis based on simulation results (Duan et al., 2023). In addition, one thing should be emphasized here is that although speleothem $\delta^{18}$O in this study is interpreted to reflect local rainfall amount, it doesn't mean that these two parameters are linearly related, especially in longer timescales. We improve our discussions about the interpretation of these proxies in the last paragraph of section 4.1.

Another issue is, in Fig. 2, the first positive shift between 8.45 and 8.39 ka BP in PC1 is more prominent than the one around 8.2 ka BP, whereas the latter excursion around the 8.2 ka BP is the most remarkable in d$^{18}$O. On the opposite, in d$^{13}$C, the average value between 8.45-8.39 ka BP is around -11‰, which is lower than before and after. If all of them are related to hydroclimatic changes, why the behaviors of the first drought are so different from these 3 proxies? Or is there any new information could be discovered from their differences.

Re: Firstly, as we demonstrate in the last review comment, the relationship between $\delta^{18}$O and trace element is necessary to be linear. Therefore, the more prominent first excursion in PC1 could reflect the impact of local precipitation that decoupled with the large scale atmospheric circulations. Secondly, the negative $\delta^{13}$C can be explained by the source of $\delta^{13}$C signal. As the interpretation of $\delta^{13}$C in section 4.1, the density of vegetation cover, the biomass activity and the vadose seepage solution play a crucial role in the variations of BH-2 $\delta^{13}$C. Since the nonlinear response of vegetation to the climate change, $\delta^{13}$C could be not always strictly follows the behavior of $\delta^{18}$O and/or trace element. In other words, the vegetation system is quite resilient to the climate change. When the climate fluctuates in a limited range or short timescale, and thus doesn't reaches a threshold value, the vegetation (i.e., $\delta^{13}$C) won't exhibit dramatic excursions. In this study, pluvial

condition at ~8.46 ka BP can be inferred from both anomalously negative $\delta^{18}O$ and trace element, which could be responsible for the coeval negative $\delta^{13}C$ values. However, the $\delta^{13}C$ didn't follow the positive trend of the other two proxies during 8.46-8.39 ka BP, which could be attributed to the nonlinear relationship between the change of vegetation and hydrological conditions, in particular during short climate excursions.

And why not use ±1 SD of the PC1 to define the drought and wet condition as well? If using the same standard adopted by the isotopes, the rebound after the 8.2 ka must not be prominent in PC1 anymore. Not mention the PC1 variations before the 8.45 ka BP are different from the isotopes, and the most prominent positive excursions of PC1 is around 8.87 ka. Sine there is no rapid rebound in d¹³C record either, if all the proxies are related to local rainfall as the authors presumed, then the rebound after the 8.2 ka in d¹⁸O is not necessary to indicate a pluvial condition.

Re: As demonstrated above, one cannot expect these three proxies strictly follow each other due to various controlling factors. Below figure shows that the similar excursions at 8.46, 8.2 ka BP, and post-8.2 ka between PC1 and $\delta^{18}O$ can be observed even we define the drought or wet conditions by ±2 SD of the PC1. Therefore, our view about the rebound after 8.2 ka event is reliable. As for the $\delta^{13}C$ record, we think the vegetation is hard to rapidly flourish from the severe damage during the 8.2 ka event, leading to relatively positive values compared to that before. Moreover, in this study, we focus on the episode after 8.5 ka BP because of the similar behavior in them, thus the most prominent excursions at around 8.87 ka BP was not discussed. According to Figure S2, this anomalous event is most attributed to the Mg/Ca ratios and not conspicuous in Ba/Ca and Sr/Ca records. But the excursions after 8.5 ka BP is common in all records. Therefore, more evidence are needed to prove the reliability. Conclusively, the discrepancy among three proxies cannot deny the pre-, at-, and post-8.2 ka events we proposed.

[Figure]

In addition, as the authors mentioned that the two-drought pattern around the 8.2 ka BP also existed in other paleo-climate records, the readers would care is there any new

information revealed from the new high-resolved multi-proxy records or is there any direct evidence which could provide new clues for the mechanism exploration?

Re: We are not sure about the meaning of 'new information'. In our opinion, the main contribution of this study is trying to prove the global significance, instead of local phenomenon, of the sequence of climate events from 8.5 to 8.0 ka BP, which was not paid enough attention before. Many records, that only discussed the 8.2 ka event before, together with our new profiles, are compiled to demonstrate the existence of the other two excursions. About the mechanism, recent studies on the marine core from the North Atlantic are synthesized to show that these climate signals presumably come from the meltwater influx into the North Atlantic and associated AMOC strength. Still, more evidences with high resolution and accurate chronology are necessary to confirm the connection between the North Atlantic and the EASM domain.

Other minor comments:

Line 84, '1998 and 2010' should be follow by 'AD' or 'CE', please correct other relevant descriptions in the rest of the manuscript.

Done

Line 84-85, better to give the summer rainfall amount as well, to show most of the local rainfall are from the summer.

Done

Line 86-89: 'termed' should be 'related'.

Done

Fig. 2a, need to put a scale on the stalagmite profile

Re: The scale of sample is the same as that in subpanel b. We add this description in the revised version and put a scale in subpanel a.

Line 92, suggest to add 'd$^{18}$O' before 'the results'.

Done

Line 101-103: the scan work belongs to the previous study (Duan et al., 2023), not to this study.

Re: The description is shortened and modified.

Line 109: 'international standards' need to be detailed with sample and No.

Done

Line 148: 'Mg/Ca' should be 'Ba/Ca'

Done

Line 151-152: Fig. 2 shows that before 8.46 ka BP, the PC1 derived from the trace elements demonstrate a general decreasing trend, which is different from the oxygen and carbon isotope variations.

Re: We add related description in this section.

Line 153, 'Aftermath' should be replaced by 'afterwards'.

Done

Line 184-197: I agree that the PC1 derived from Mg/Ca, Sr/Ca and Ba/Ca is mainly influenced by PCP. But it should be noticed that PCP occurs under dry conditions, so trace element ratios are more sensitively to drought, but not so sensitively to wet condition.

Re: We thank the reviewer for this suggestion and agree with it. This is the reason that trace element ratios can be used to indicate local wetness conditions. Under dry conditions, PCP occurs in the path way of solution and causes higher trace element ratios (i.e., PC1 values). In contrast, when hydroclimate is wet, sufficient precipitation supply can suppress the PCP processes, and thus more Ca element is preserved in solution to form lower trace element ratios.

Line 206-208: 'One noticeable feature of our δ18O record is a switch from relatively muted to highly variable episodes divided at ~8.5 ka BP, consistent with the absence and dominance of centennial to inter-decadal periodicity before and after 8.5 ka BP, respectively (Figure 2).' Why not carry on a periodicity analysis? And what cause the periodicity change before and after 8.5 ka BP?

Re: We thank the reviewer for this suggestion. The wavelet periodicity analysis result is added to Figure 2. As can be seen, the centennial to interdecadal periodicity is prominent after ~8.5 ka BP but almost absent before this time, consistent to what we demonstrated. About the mechanism to cause this phenomenon, we propose that in the background of overall strengthened ASM during 9.0-8.0 ka BP, a series of abnormal climate events originating from the high north latitudes lead to relatively more frequent high-amplitude oscillations in $\delta^{18}O$ profiles and hence more prominent periodicity after 8.5 ka BP.

Line 226, 'concert' should be 'concern'.

Re: We didn't change this because we think the phrase 'in concert with' means 'in agreement with', therefore is more suitable than 'in concern with' which tends to give a meaning of 'considering'.

Line 282, 'because high northern latitude climate variations can strongly affect the westerly changes and finally influence the EASM', 'affect' and 'influence' should be 'be affected' and 'be influenced'.

Re: We can't agree with this opinion. The anomalous climate signals in this study could originate from high latitudes and was transmitted to EASM through teleconnection, instead of the opposite processes.

Line 307-308, 'Intriguingly, speleothem record from Padre Cave (Cheng et al., 2009) fails to preserve as clear pre- and post- 8.2 ka events as its adjacent Lapa Grand Cave (Figure 4), presumably due to different cave settings.' This could not convince people.

We modify this sentence.

Line 319,need to give the whole name of the abbreviation of LAO

Done

Line 632-646: suggest to merge Fig. 3 and Fig. 4

Done

FigS1, the background is too dark, and it is difficult to figure out the locations. The d¹⁸O comparison figure is bit a mess. What is the purpose of the comparison?

Re: We modify this figure. Actually, we didn't mean to compare these data to give some conclusions, but collected the data we mentioned in our main text as many as possible.

---

## Author Comment (AC3)

Duan et al. provide a new high resolution 8.2ka speleothem record spanning 9.0-7.9ka BP period in north China. They reconstructed the time series over this period using a variety of indicators, including carbon and oxygen isotopes, ratios of trace elements, and growth rate. The authors identified two drought periods of 8.4 and 8.2ka and found that the behavior of carbon and oxygen isotopes and trace elements was different, which may be responsible for the nonlinear response of the local ecosystem. They suggest that there were several centennial scale climate fluctuations around the 8.2ka event, and that two droughts-one pluvial pattern between 8.5-8.0ka were a widespread event on a global scale and were closely related to the northern high latitude. The manuscript is carefully prepared, this record is relatively rare in northern China, and the age model is accurate, the resolution is super high, after a certain amount of thought and revision this paper is recommended for publication.

Re: We thank the reviewer for the positive comments.

**General notes**

1. Cheng et al.(2009) and your last paper( Duan et al.,2023) reported that the 8.2ka event had a two-stage structure, but this manuscript and Tan et al.(2020) believe that there is a two drought-one pluvial pattern at 8.2ka and 8.4ka, does this similar expression give rise to some misunderstanding? Because in some papers 8.4ka and 8.2ka together constitute the 8.2ka cold event.

   Re: We clarify here that the two drought-one pluvial structure in this study is different from Cheng et al. (2009) and Duan et al. (2023). Specifically, papers of Cheng et al. and Duan et al. mainly focused on the 8.2 ka event that lasts from ~8.30 to 8.10 ka BP and revealed a two drought-one pluvial pattern within this event. Our study suggests that, in addition to the 8.2 ka event, another multidecadal drought event should occur at ~8.4 ka BP. As for Tan et al. (2020), we believe that the speleothem records from Northwest China support the opinion of more than one drought events during 8.5-8.0 ka BP, despite the lack of resolution and accurate age control.

2. What are the periodic changes of trace elements, oxygen isotopes and carbon isotopes? Do they have a common period and are they influenced by Solar Output, AMO, PDO or even ENSO on a short time-scale?

   Re: The wavelet analysis of carbon and oxygen isotope timeseries was conduct and the results are shown below. The 10% significance level against red noise is shown as a thick contour. As can be seen, two profiles don't show a common period because the $\delta^{18}O$ is dominated by interdecadal to centennial periodicity after the 8.5 ka BP while the $\delta^{13}C$ is dominated by 10~30 year periodicity almost throughout the entire study interval. To investigate the impact of the solar output, AMO, PDO and ENSO on speleothem proxies, the signals of AMOC variability should be removed, which is beyond the scope of this study. Even so, we will try to figure out this issue in next step as the reviewer suggests.

[Figure]

3. Can the $\delta^{234}U_{initial}$ attached to the dating results be used as an indicator to reflect these climatic events? Can the carbon isotope at 8.4ka be considered a wet event?

Re: Speleothem $\delta^{234}U_{initial}$ is a complex indicator to investigate the local hydroclimate change. The comparison results between $\delta^{234}U_{initial}$ (the lowest curve) and other proxies are shown in below figure. As can be seen, the $\delta^{234}U_{initial}$ values reach the highest at ~ 8.4 ka BP and the lowest at ~8.2 ka BP, in contrast to the consistently positive shifts in the other proxies. This indicating that complicated mechanisms exert influence on the speleothem $\delta^{234}U_{initial}$ signal. Moreover, resolution of the $\delta^{234}U_{initial}$ record is low, possibly limiting the detailed comparison with other proxies.

We don't think the 8.4 ka event can be considered a wet event. The succession of vegetation system is quite resilient to the climate change, and thus $\delta^{13}C$ values of speleothem not always follow the patterns of $\delta^{18}O$ and trace element ratios. In this study, pluvial conditions at ~8.46 ka BP can be inferred from both anomalously negative $\delta^{18}O$ and trace element ratios, which could be responsible for the simultaneous negative $\delta^{13}C$ values. However, the $\delta^{13}C$ don't follow the dramatic positive excursion of the other two proxies during 8.46-8.39 ka BP, which could be attributed to the nonlinear relationship between the change of vegetation and hydrological conditions, in particular during short climate excursions. On the other hand, a relatively stable plant community and well-developed soil could be formed above the karst zone at that time, which increase the resilience of vegetation to environmental variations during and just after pluvial period, in turn suppressing the large and rapid variation of $\delta^{13}C$ in the karst system. But still, an inconspicuous positive trend can be noticed in the carbon isotope timeseries in this interval, especially the highest value up to ~-9.1 ‰ at 8.40 ka BP.

[Figure]

4. Carbon isotopes vary much more than oxygen isotopes, as is the case in Hulu cave. It is generally assumed that carbon isotopes are more sensitive to climate change. Why is there a nonlinear response now?

Re: As our response to Comment 3, local vegetation system are somehow resilient to climate variation when the tolerance threshold of vegetation system is not broken. Therefore, delay and/or smoothed signals are observed in $\delta^{13}C$ relative to the $\delta^{18}O$ records, especially in short time scales and/or limited range of climate change. In longer time scale, however, speleothem $\delta^{13}C$, as a rainfall amount proxy, in response to climate change could be more sensitive than $\delta^{18}O$ as demonstrated by Li Yunxia et al. (2020, EPSL). Therefore, there is no conflicts between this study and previous studies on the explanation of the carbon isotope considering the local environment of each cave location and time scale.

**Specific notes**

1. Figure2a, this paper is based on Duan et al., 2023 (GRL) to add new data for research. Please distinguish the ages used in the previous paper and the new ages added in this paper in this figure. Figure2f, from 9.0-8.5ka, there is a clear decreasing trend in the PC1 index of trace elements in this period, indicating that the climate is becoming wetter, which is not reflected in oxygen and carbon isotopes.

Re: We change the published age results to red color in Figure 2b. It seems that the decreasing trend in the PC1 arises from the anomalously positive Mg/Ca excursion at ~8.87 ka BP, without which this trend is much flat as can be seen in separated trace element ratio records (Figure S2).

[Figure]

2. Figure3, in addition to the dating error of LAO you have marked, the marking of the dating error bar of other stalagmites records is conducive to the understanding of this paper.

    Re: Done.

3. Figure s2, what are the correlation between the three trace elements, if there are calculations of the correlation, you can show their consistency.

    Re: The correlation coefficients for Mg/Ca and Sr/Ca, Mg/Ca and Ba/Ca, and Sr/Ca and Ba/Ca, are 0.24 ($p<0.01$), 0.49 ($p<0.01$), and 0.47 ($p<0.01$), respectively.

---

## Author Comment (AC4)

Duan et al. present a new multiproxy speleothem record of East Asian summer monsoon hydroclimate variability in northern China during the 8.2 ka event. The record is of high resolution, has a robust age model, contains multiple proxies, and is well interpreted. The scientific questions being asked are relevant to the journal, and the paper is overall well written. Thus, I recommend the manuscript for publication and have only minor comments to add to previous reviews:

I recommend the authors providing scatter plots and/or correlation matrix showing the relationship and accompanying statistics between the trace elements and stable isotopes.

Re: We thank the reviewer for the positive comments and constructive suggestions on our work. Unfortunately, the Pearson correlation results show that there is no significant and strong relationship ($r<0.3$, $p>0.2$) between the trace elements and stable isotopes, although the visual inspection suggests conspicuous excursions at ~8.4 and 8.2 ka BP in all records. Intriguingly, when we focus on the interval of 8.5-8.0 ka BP, the correlations become significant ($p<0.05$) and the coefficients are higher than 0.3, except the Ba/Ca ratio ($p>0.1$). This could be attributed to the short duration of climate events relative to the entire record, presumably pointing to the complex local controlling factors when the external climate pressure was not at play.

Might be worth plotting up and citing the recent paper by Wood et al. (2024) that documents the 8.2 ka event in northern Laos (Wood, Christopher T., Kathleen R. Johnson, Lindsey E. Lewis, Kevin Wright, Jessica K. Wang, Andrea Borsato, Michael L. Griffiths et al. "High-Resolution, Multiproxy Speleothem Record of the 8.2 ka Event From Mainland Southeast Asia."*Paleoceanography and Paleoclimatology* 38, no. 12 (2023): e2023PA004675). This record also shows no sign of an earlier (i.e., ~8.4 ka) drought event, which may provide further support for their hypothesis that this precursor event was restricted to the higher latitudes.

Re: Wood et al provide an unprecedented multiproxy 8.2 ka record based on speleothem from Mainland Southeast Asia. They reveal a dry 8.2 ka event for the first time and possible pluvial precursor event, whereas the dry 8.4 ka event unraveled by this study is absent. In our opinion, whether the 8.4 ka event is recorded or not is not dependent on the latitudes, but the climate sensitivity of archive site. For example, Huangyuan and Wuya caves are located in the margin of Asian summer monsoon, and Hoti cave is situated in the margin of Indian summer monsoon (23°5′N). In contrast, the record of Wood et al is from the core region of Intertropical convergence zone belt (1195 mm annual precipitation), thereby the weak climatic signal from the high latitudes during the 8.4 ka event could have been covered by local factors. We cited this work in the discussion section.

I am curious as to the timing of the 8.2 ka event in the d18O using different age model algorithms. For instance, OxCal shows a very constant growth rate, particularly during the earlier part of the record, that doesn't appear to be matched by the U-Th dates—i.e., the dates suggest more variability in the growth rate than is observed in the age model. Also, beginning at a depth of ~70 mm, the age model passes through the confidence intervals of

most dates rather than the actual date. Is this also the case if another algorithm is used instead (e.g., COPRA, StalAge etc). There was also no mention of the methods for layer counting and the age model derived from this method.

Re: We thank the reviewer's concerns about the chronology reconstruction and growth rate.

We compare the growth rate results based on U-Th dates and Stalage and Oxcal age models. As can be seen, the growth rate is more variable, especially in the stage before 8.5 ka BP, in U-Th age and Stalage modelled result, whereas is smoothed in the other age model. The smoothed curve could result from the algorithm of age model, which is based on the principle that the growth rate of speleothem unlikely changes within short time. To produce a fitting curve, the growth rate among contiguous U-Th ages is smoothed within or even sometime a little beyond the dating errors. It seems that the chronology established by Stalage program is more suitable relative to the Oxcal method because the former one has fewer out-of-confidence interval U-Th ages. Therefore, we replace the BH-2 records established by the Stalage age model in the revised manuscript. However, it is worth emphasizing that the growth rates based on all three methods consistently display lower excursion between ~8.5 and 8.1 ka BP, not contradict with our view in the manuscript that the drought conditions, indicated by stable isotopes and trace elements, induced slow growth rate of speleothem. In other words, the choice of reconstruction method of growth rate does not affect our conclusions of a series of climate events during 8.5-8.0 ka BP rather than only one. Even though the possible uncertainty on the accurate age for the 8.4/8.5 ka event, we are quite confident with our 8.2 ka event because of the negligible offset in comparison with floating chronologies for 8.324–8.077 ka BP and [230]Th dates within uncertainties.

The results of layer counting and relative age model methods were cited from the published paper by Duan et al. (2023), and we didn't revise the data in this study. (*Duan, P., Li, H., Ma, Z., Zhao, J., Dong, X., Sinha, A., et al. (2023). Interdecadal to centennial climate variability surrounding the 8.2 ka event in North China revealed through an annually resolved speleothem record from Beijing. Geophysical Research Letters, 50, e2022GL101182. https://doi.org/10.1029/2022GL101182*). In brief, the least square method was used to establish the chronology through anchoring annual lamina counting to the encompassed seven [230]Th dates in the speleothem section of 15-43 mm. To establish the consecutive chronology for the entire record, all above fitting results for each lamina in 16–43 mm (corresponding to 8.077–8.324 ka BP) with uncertainties and the other [230]Th dates in the remnant study section are input to age model algorithm. The out-of-confidence interval at ~70 mm could be the smoothed result in order to fitting the massive age constrains at 15-43mm.

[Figure]

Additional methods on the Laser Induced Breakdown Spectroscopy (LIBS) are needed. For example, what standards were used for calibration and correction for instrumental drift? What were the RSDs for standards?

Re: We provide more information about our LIBS method. "Trace element ratios (Mg/Ca, Sr/Ca, Ba/Ca), of which the intensity ratio of emission lines are 285.2 (Mg), 407.8 (Sr), and 493.4 (Ba) nm relative to 373.7 nm (Ca), were measured using Laser Induced Breakdown Spectroscopy (LIBS) following the detailed description in Li et al. (2018). In brief, analyses were performed by pulsing and focusing yttrium-aluminum-garnet-Nd laser beam to 0.1 mm. Emitted plasma from the stalagmite surface was collected by optical fibers and sent to a four-passage spectrometer (Ocean Optics MX500+) to obtain a spectrum within the 200-to 580-nm range. These data were determined through the intensity of characteristic spectral line for each element, and then the intensity ratio of each trace element signal to Ca element was calculated and output as the final result for each point. The obtained record is the median intensity ratio based on 20 pulses at each sampling site after 5 laser shots for pre-cleaning the surface. The measurements were performed continuously along the speleothem's growth axis at 0.3 mm increment and a total of 565 data were obtained. The accuracy of data was ensured through the excellent replicability between two-time measurements instead of inset standard materials because of the overwhelming amount of Ca relative to trace elements in speleothem. The original spectral data were processed using an interface created in MATLAB (2020a). The typical standard deviation for the average signal intensity is less than 0.02 (without unit)."

As can be seen in below figure, trace element ratio records between two-time measurements are broadly similar despite of the discrepancy of absolute values in some points, suggesting the stability of LIBS device and measuring approach.

[Figure]

Line 268: "*presumably were definitely correlated on a broad pattern but did not necessarily exactly follow each other.*" This needs to be quantified as stated above.

Re: please see the response to the first comment.

---

## Author Response (AR2)

Dear authors,

Thanks for your revised ms. I am sorry to hold this up, but could you please provide more information on exactly how you derived the composite age model in Figure 2b? It is not easy to understand the procedure based on the information provided in the figure caption, and in any case a full explanation should be provided in the main text (the Methods section preferably). For example, I am guessing that StalAge was applied to all of the U-Th ages/depths, but it not clear from the information provided how you incorporated/spliced the layer-counted section to the U-Th-based parts either side. I realise this may have minimal impact on your final age estimates of key events, etc. but I am sure that (technically) how you carry out this anchoring will be of interest to readers, especially those specialising in speleothems.

Re: We thank the editor for the suggestions. In the revised version, we added the relative description about the chronology reconstruction using the combination of layer band counting and Stalage algorithm. In detail, the fitted age and error for each annual band between 16 and 43 mm were obtained based on the least square method (Duan et al., 2023). To establish a consecutively composite chronology for the entire record, all these fitting results in 16–43 mm (corresponding to 8.077–8.324 ka BP) with uncertainties and the other fifteen [230]Th dates in the remnant study section were input to Stalage algorithm. In this way, the seven [230]Th dates drilled from 16–43 mm were only used in the layer band counting procedure but not the Stalage age model. The output results of Stalage were adopted as the reconstructed chronology for isotope and trace element records.

Please note also that: (i) the 'layer counting' label in Fig 2b has a spelling error, which should be fixed; (ii) please make the final age model line used in the paper more obvious, as it is difficult to see resolve in the PDF; (iii) one of the red ages from Duan et al. (2023) seems to be a new age for this paper (the youngest age), as it doesn't appear in table S1 of the 2023 paper; and (iv) the second oldest age (BH-2-6) from Duan et al. (2023) has a different depth in this current manuscript - if this has been updated from the previous paper, please let the reader know, but if not, please correct it.

Re: i) the spelling error was corrected; ii) the final age model line was replaced with bold blue line; iii) the youngest red age is a new one present in this study and hence should be marked by black; iv) we corrected this depth from 43mm to 45 mm.

Note: we rearranged the information of foundation grants in the acknowledge section.

---

## Author Response (AR3)

Add: No. 28 Xianning West Road, Xi'an,
Shaanxi, P.R. China, Zip: 710049
http://www.xjtu.edu.cn

May 13, 2024

Dear Editor,

This letter accompanies our revision of manuscript entitled "**A series of climate oscillations around 8.2 ka BP revealed through multi-proxy speleothem records from North China**".

We sincerely thank you for accepting our manuscript. Since there is no more suggestion or comments, we only double-checked the manuscript in this stage except deleting the description of "when the manuscript is accepted" in the data availability section.

Sincerely,

Prof. Hai Cheng and Dr. Hanying Li
Institute of Global Environmental Change, Xi'an Jiaotong University
Xi'an 710049, China
E-mail: cheng021@xjtu.edu.cn and hanyingli@xjtu.edu.cn